# Which Adverse Events and Which Drugs Are Implicated in Drug-Related Hospital Admissions? A Systematic Review and Meta-Analysis

**DOI:** 10.3390/jcm12041320

**Published:** 2023-02-07

**Authors:** Annette Haerdtlein, Elisabeth Debold, Marietta Rottenkolber, Anna Maria Boehmer, Yvonne Marina Pudritz, Faiza Shahid, Jochen Gensichen, Tobias Dreischulte

**Affiliations:** 1Institute of General Practice and Family Medicine, University Hospital, LMU Munich, 80336 Munich, Germany; 2Doctoral Program Clinical Pharmacy, University Hospital, LMU Munich, 81377 Munich, Germany; 3Institute of Pharmacy, Department of Clinical Pharmacy, University of Bonn, 53121 Bonn, Germany; 4Hospital Pharmacy, University Hospital, LMU Munich, 81377 Munich, Germany

**Keywords:** adverse drug event, adverse drug reaction, emergency department, hospital admission, hospitalization, meta-analysis, systematic review

## Abstract

Adverse drug events (ADEs) and adverse drug reactions (ADRs) are leading causes of iatrogenic injury, which can result in emergency department (ED) visits or admissions to inpatient wards. The aim of this systematic review and meta-analysis was to provide up-to-date estimates of the prevalence of (preventable) drug-related ED visits and hospital admissions, as well as the type and prevalence of implicated ADRs/ADEs and drugs. A literature search of studies published between January 2012 and December 2021 was performed in PubMed, Medline, EMBASE, Cochrane Library, and Web of Science. Retrospective and prospective observational studies investigating acute admissions to EDs or inpatient wards due to ADRs or ADEs in the general population were included. Meta-analyses of prevalence rates were conducted using generalized linear mixed models (GLMM) with the random-effect method. Seventeen studies reporting ADRs and/or ADEs were eligible for inclusion. The prevalence rates of ADR- and ADE-related admissions to EDs or inpatient wards were estimated at 8.3% ([95% CI, 6.4–10.7%]) and 13.9% ([95% CI, 8.1–22.8%]), respectively, of which almost half (ADRs: 44.7% [95% CI: 28.1; 62.4]) and more than two thirds (ADEs: 71.0% [95% CI, 65.9–75.6%]) had been classified as at least possibly preventable. The ADR categories most frequently implicated in ADR-related admissions were gastrointestinal disorders, electrolyte disturbances, bleeding events, and renal and urinary disorders. Nervous system drugs were found to be the most commonly implicated drug groups, followed by cardiovascular and antithrombotic agents. Our findings demonstrate that ADR-related admissions to EDs and inpatient wards still represent a major and often preventable health care problem. In comparison to previous systematic reviews, cardiovascular and antithrombotic drugs remain common causes of drug-related admissions, while nervous system drugs appear to have become more commonly implicated. These developments may be considered in future efforts to improve medication safety in primary care.

## 1. Introduction

Drug therapy plays a major role in the prevention and treatment of many illnesses and is the most common intervention in primary care. However, adverse drug events (ADEs) and adverse drug reactions (ADRs) are also leading causes of iatrogenic injury and avoidable harm and costs [1,2]. The World Health Organization (WHO) and national governments have therefore declared the prevention and reduction in medication-related harm public health priorities [1,3].

Drug-related emergency department (ED) visits and hospital admissions can be seen as the tip of the iceberg of drug-related harm originating in primary care and as such have been extensively studied over the last two decades [4,5,6,7,8]. Systematic reviews estimate that between 5% and 30% of ED visits and hospital admissions are attributable to ADRs and ADEs [7,9,10,11]. However, prevalence estimates vary by a number of factors, including for example, the study population and the definitions of ADEs and ADRs applied [12,13,14]. Nevertheless, ADEs are generally seen as a broader concept than ADRs, and additionally include adverse events that are temporarily associated with (but not necessarily caused by) drugs as well as adverse events attributable to overdosing and underuse of drug therapy [15,16].

Despite the large body of previous research on the subject, studying the scale of drug-related hospital admissions remains important, both to inform interventions and to evaluate their impact. In addition, it is vital to understand which drugs and adverse events are the main contributors to drug-related admissions. Even though some root causes of preventable drug-related admissions may be generic [6], drug- and/or adverse event-specific solutions may be required to prevent them. Periodic updates to estimate the scale and describe the nature of drug-related admissions are important in order to account for the possible impact of interventions to improve medication safety as well as new drugs entering the market.

The aim of this systematic review and meta-analysis was therefore to update estimates of (1) the prevalence of (preventable) drug-related ED visits and hospital admissions, as well as the (2) adverse events and (3) drugs predominantly implicated.

## 2. Materials and Methods

### 2.1. Study Design and Reporting

We conducted a systematic review and meta-analysis of observational studies published over the last ten years (2012–2021), which examined the prevalence and/or the causes of drug-related hospital admissions and ED visits in the general adult population. For meta-analyses, the primary endpoint of interest were ADR-related admissions to EDs or inpatient wards for all three research aims. We also examined the prevalence of ADE-related admissions to EDs or inpatient wards and the proportions of preventable ADRs and ADEs as secondary endpoints. The systematic review is reported according to the Preferred Reporting Items for Systematic Reviews and Meta-Analyses (PRISMA) statement [17] (see Appendix A).

### 2.2. Literature Search

An electronic literature search of studies published between January 2012 and December 2021 was performed in the following databases: PubMed (by title/abstract), EMBASE, Cochrane Library, and Web of Science (each by titles only). The applied search strategy combined the outcome focus (ADRs or ADEs) with the setting of hospital admission. The detailed search strategy is provided in Appendix A.

### 2.3. Study Selection Process

Two reviewers independently screened all identified titles and abstracts for eligibility and assessed the full texts of eligible studies. The two reviewers first discussed disagreements amongst them and consulted a third reviewer where they could not be resolved.

### 2.4. Eligibility Criteria

*Eligible studies*—We included observational studies with prospective or retrospective data collection investigating acute admissions to EDs or inpatient wards due to ADRs or ADEs in the general adult population. Eligible studies had to report outcomes relevant to at least one research aim. Only studies that applied causality assessment to medical charts of individual cases using explicit algorithms [18,19] or a comparable standardized process were included. Studies were eligible if they were published in English or German irrespective of their geographic origin.

Since our aim was to estimate ADR prevalence in the general adult population, we excluded studies, which exclusively focused on patients in specific age groups (e.g., only pediatric or older patients) or admitted to specific wards, those with specific clinical diagnoses or those taking particular drug classes. We also excluded studies, which focused solely on specific ADRs (e.g., due to drug poisoning or drug abuse) and those analyzing spontaneous ADR reporting data bases or administrative datasets, as they are known to underestimate ADR prevalence [12,13]. Reviews, newspaper articles, expert opinions, commentaries, discussion papers, journalistic interviews, policy reports, books, and conference abstracts were also excluded.

*Eligible drug-related admissions*—In order to be considered as drug-related admissions to EDs or inpatient wards, we stipulated a) that ADRs or ADEs (i.e., the relationship between an adverse event and drug therapy) had to be characterized by causality assessments as either ‘possible’, ‘probable’ or ‘certain’ and b) that ADRs or ADEs had to be characterized as either causing or contributing to the hospital admission. Adverse events with unlikely/doubtful causal relationships to drug therapy, including those described as simply occurring on admission were therefore excluded.

### 2.5. Assessing Reporting Quality and Risk of Bias

The reporting quality of the included studies was assessed by using the Strengthening the Reporting of Observational Studies in Epidemiology (STROBE) criteria for observational studies [20]. We used 25 items that we considered relevant to our research questions and assigned each item 1 point when the criteria were completely met, and 0 points otherwise. For each study, we calculated the overall reporting quality as the proportion (%) of 25 points achieved.

In addition, we examined all included studies for 12 specifically defined criteria representing potential sources of heterogeneity and implying either a risk of selection (sampling) bias (i.e., the sampling of research sites and cases; 5 items) or misclassification bias (i.e., methods used to identify and verify the presence of drug-related admissions; 7 items). These included aspects recently identified by Wallerstedt et al. as potentially contributing to the diverse prevalence rates of drug-related admissions reported in the scientific literature [14].

Both the reporting quality and risk of bias were assessed by one reviewer and checked by a second reviewer. Uncertainties were discussed and resolved within the research team.

### 2.6. Data Extraction

A custom Excel template was used to extract all relevant data from the eligible studies. Two reviewers independently extracted the data and discussed disagreements prior to consulting a third reviewer to resolve them where necessary. The extracted data were based on information reported in or calculated from the included articles. Extracted data included (i) basic study characteristics (including, e.g., study design, number of centers, study population), (ii) characterization of drug-related admissions (including ADR/ADE definitions and detection methods), (iii) prevalence of ADR/ADE-related hospital admissions including any reported results of causality and preventability assessments, (iv) outcomes (i.e., prevalence of detected ADRs/ADEs, as well as associated drugs and drug classes).

### 2.7. Statistical Analysis

#### 2.7.1. Prevalence Estimates in Each Study

For each individual study, the prevalence of drug-related admissions to EDs or inpatient wards (research aim 1) was calculated as the number of cases admitted to EDs or inpatient wards due to at least one ADR or ADE (numerator) divided by the total number of cases included and assessed for ADRs/ADEs during the study period (denominator).

Some studies reported implicated ADRs (research aim 2) and drugs (research aim 3) as proportions of all patient cases (type A prevalence rates), whereas others reported them as proportions of all implicated ADRs and drugs (type B prevalence rates). Since more than one ADR may be involved in each patient case, these prevalence rates are not identical, and we therefore analyzed them separately. We were primarily interested in type A prevalence rates and declared them our primary endpoint, while type B prevalence rates were considered secondary outcomes.

For each point estimate, we calculated a 95% confidence interval (CI) using the Clopper–Pearson method [21]. We calculated the prevalence rates of preventable ADRs/ADEs as the proportions of drug-related admissions/ED visits (denominator), where preventability was reported as either *possible*, *preventable*, or *certain*.

#### 2.7.2. Meta-Analyses of Prevalence for Research Aims 1 to 3

To meta-analyze the prevalence of admissions to EDs or inpatient wards (research aim 1), we conducted separate analyses for admissions due to (1) ADRs and (2) ADEs.

To meta-analyze the proportions of implicated ADRs (research aim 2), we iteratively developed a coding frame (with two hierarchical levels) based on the International Classification of Diseases version 10 (ICD-10) and the Medical Dictionary for Regulatory Activities (MedDRA) [22,23]. Reported ADRs that did not use either of these coding systems were assigned to the most suitable ADR category.

To meta-analyze the proportions of implicated drugs (research aim 3), the reported drugs and drug classes were coded according to the Anatomical Therapeutic Chemical (ATC) classification system [24] using four hierarchical levels.

ADRs and drug groups were generally only included in meta-analyses at the hierarchical levels by which they were reported in the original studies [24]. For example, events reported as *hypotension* or *bradycardia* were only included in meta-analyses at these levels. The same applies to suspected drug groups, such as *high-ceiling diuretics.* Only when individual study reports included comprehensive listings of all ADRs and drugs, these events contributed to meta-analyses at higher hierarchical levels (e.g., *cardiovascular disorders* or *diuretics*, respectively), as in these cases there is no risk of (under)reporting bias.

For all meta-analyses, we used generalized linear mixed models (GLMM) with the random-effect method [25]. The “meta” package (version 5.2-0) of the statistical software package R (version 4.1.3) was used for pooling. Forest plots were utilized for graphical representation of the pooled prevalence. Statistical heterogeneity for the group of studies was analyzed using the I^2^ statistic. An I^2^ value > 25%, >50%, and >75% was considered to represent low, moderate, and high heterogeneity, respectively [26].

#### 2.7.3. Sensitivity Analyses

In order to explore residual heterogeneity between studies, we conducted a number of sensitivity analyses: In sensitivity analysis A, we restricted the main analysis to studies which had clearly excluded ED visits that did not result in admission to inpatient wards, and in sensitivity analysis B, we further restricted analysis to hospital admissions that were *caused* by ADRs (i.e., excluded those where ADRs only *contributed* to admissions). In sensitivity analysis C, we excluded studies from the main analysis using trigger tools to first screen admissions for the presence of ADRs or ADEs prior to conducting causality assessments.

## 3. Results

### 3.1. Literature Search and Study Selection

The literature search identified 1280 citations as shown in the PRISMA flow chart (Figure 1). After screening of titles and abstracts, the full texts of 130 articles were reviewed. Of these, 112 were excluded, most commonly due to their focus on specific patient populations or clinical wards. Finally, 17 studies reported in 18 articles satisfied our eligibility criteria and were included.

### 3.2. Study Characteristics

Table 1 presents details of the 17 included studies, which were published between 2013 and 2021. Most of the studies were conducted in Europe (*n* = 9) [5,27,28,29,30,31,32,33,34,35], with a further five studies from Asia [36,37,38,39,40], two from Australia [41,42], and one from North America (Canada) [43]. Average age and gender distribution were comparable across all studies. However, the included studies differed in terms of a number of other methodological considerations, including (a) whether the data collection was prospective (*n =* 13) or retrospective (*n =* 4); (b) whether only admissions to inpatient wards were considered (*n =* 11) or also emergency department (ED) visits (*n =* 6); (c) whether only ADRs were reported (*n =* 12), only ADEs (*n =* 3) or both (*n =* 2); (d) which ADR or ADE definition was used (for more detailed information, see Appendix A); (e) which causality assessment method was used (Naranjo [18]: *n =* 7, WHO-UMC [19]: *n =* 5, Spanish Pharmacovigilance System [44]: *n =* 2, others [45,46,47]: *n =* 4); (f) which preventability assessment method was used (Hallas criteria [48]: *n =* 5, Schumock and Thornton [49]: *n =* 2, others [50,51]: *n =* 2).

### 3.3. Quality of Reporting

Of 25 items based on the STROBE criteria (with 25 maximum achievable points), the median (interquartile range (IQR)) score achieved was 72% (68%; 80%) with seven studies scoring > 75%, and only one study scoring below 50%. In 14 (82.4%) of 17 studies, the limitations stated were judged as adequate. However, efforts to address potential sources of bias were only reported in seven (41.2%) studies, and even fewer (4/17 (23.5%)) stated how often data were missing or how the data were handled. Further details are provided in Appendix A.

### 3.4. Risk of Bias

*Selection bias.* Of 17 included studies, 14 (82.4%) were monocentric. Five studies (29.4%) excluded patients who were unwilling or unable to consent to participation. In six studies, patient recruitment was restricted to working days or on-call days. In two studies (11.8%), patients were screened using a trigger tool (i.e., a pre-defined list of diseases or syndromes potentially caused by drugs) and those with a negative screening result were excluded.

*Misclassification bias.* Nine (52.9%) studies explicitly stated which ADR/ADE definition they used, and only three (17.6%) assessed whether ADRs were the reason for admission or merely contributed to it. For causality assessment, six different methods were used. In eight of twelve studies reporting the findings of causality assessments, the majority of ADRs were classified as ‘possible’. Two studies (11.8%) only considered ‘certain’ and ‘probable’ ADRs for further analysis. In 13 (76.5%) studies, causality assessment was conducted by more than one investigator. Inter-rater agreement was reported in five studies (29.4%) and varied from slight to almost perfect agreement. Further details are provided in Appendix A.

### 3.5. Prevalence of Drug-Related Admissions

In 15 of the 17 studies (88.2%), the prevalence of admissions to EDs or inpatient wards due to ADRs or ADEs was reported or could be calculated. In the remaining two studies, the source population (denominator) was either not specified [33,34] or unclear (Table 1) [36].

*ADR-related admissions.* The reported prevalence rates for ADR-related admissions to EDs or inpatient wards ranged from 4.2% (186/4403) to 30.0% (9/30) with an overall median of 8.2% (interquartile range (IQR) 6.1–9.2%). Figure 2 shows that the meta-analysis of 12 studies found a prevalence rate of ADR-related admissions to EDs or inpatient wards of 8.3% (95% CI, 6.4–10.7%], with the I^2^ statistic (I^2^ = 95%, *p* value < 0.01) showing considerable heterogeneity.

*ADE-related admissions.* The reported prevalence rates for ADE-related admissions to EDs or inpatient wards ranged from 4.9% (198/4020) to 30.7% (133/434). Figure 3 shows that the meta-analysis of five studies found a prevalence rate of ADE-related admissions to EDs or inpatient wards of 13.9% ([95% CI, 8.1–22.8%]), with the I^2^ statistic (I^2^ = 99%, *p* value < 0.01) again showing considerable heterogeneity.

*Sensitivity analysis.* All three sensitivity analyses yielded findings broadly consistent with the main analysis. When we restricted analysis to ADRs/ADEs resulting in admission to inpatient wards in sensitivity analysis A, the prevalence for ADRs was higher at 9.3% (95% CI, 6.4–13.3%) and lower for ADEs at 12.0% (95% CI, 5.8–23.1%). When we further restricted analysis to admissions to inpatient wards that were *caused* by ADRs or ADEs (rather than just *contributing* to them) in sensitivity analysis B, the prevalences for both ADRs and ADEs were lower at 7.3% (95% CI, 5.7–9.5%) and 8.9% (95% CI, 3.8–19.3%), respectively. When we excluded studies from the main analysis using trigger tools in sensitivity analysis **C**, the ADR prevalence was slightly higher at 8.9% (95% CI, 6.8–11.6%), whereas none of the ADE studies used trigger tools. Forest plots for the sensitivity analyses are provided in Appendix A.

### 3.6. Preventability of ADRs/ADEs

Five studies found ADR preventability rates ranging from 24.0% to 97.6%, while three studies found ADE preventability rates ranging from 65.3% to 73.7% (Table 1).

*Preventability of ADR-related admissions*. Figure 4 shows that the meta-analysis of four studies found an ADR preventability rate of 44.7% (95% CI, 28.1–62.4%) with high heterogeneity across studies (I^2^ = 94%, *p* value < 0.01).

*Preventability of ADE-related admissions*. Figure 5 shows that the meta-analysis of two studies found a preventability rate of 71.0% (95% CI, 65.9–75.6%) with low heterogeneity (I^2^ = 44%, *p* value < 0.01).

### 3.7. Implicated ADRs

For the analysis of implicated adverse events and drugs, we exclusively considered studies which reported on ADRs. Of 14 studies, 9 (64.3%) were included in the analyses, with the other 5 studies either providing no [37,40] or no exact [5] numbers on ADR/drug frequencies, or reporting proportions with denominators that were incompatible with our analyses [29,31].

Of the seven studies that provided the numbers of implicated ADRs, four reported prevalence rates of implicated ADRs as proportions of patient cases with ADR-related admissions (*type A* prevalence rates) [28,32,33,43], and five studies reported them as proportions of all ADRs (*type B* prevalence rates) [28,30,38,41,43], i.e., from two studies both prevalence types could be extracted [28,43]. The list containing all ADR frequencies is provided in Appendix A.

Our iteratively developed ADR coding frame distinguishes 19 ADR categories and 56 individual ADRs. Table 2 ranks type A prevalence rates of ADR categories at the highest level in descending order of estimated point prevalence rates. Full details of lower hierarchical levels and type B prevalence rates are provided in Appendix A.

Four ADR categories had type A prevalence rates of ≥ 10.0% in meta-analysis, where gastrointestinal disorders were the most common (28.5 [95% CI: 21.6; 36.6]%), followed by electrolyte disturbances (16.5 [9.5; 25.7]%), bleeding events (13.5 [9.5; 18.8]%), and renal and urinary disorders (11.8 [5.4; 24.1]%). All of these also accounted for ≥ 10.0% of all ADRs in studies reporting type B prevalence rates and no further ADR was found to have a type B prevalence rate of ≥10.0%.

### 3.8. Implicated Drugs and Drug Groups

Of eight studies that reported prevalence rates of implicated drugs or drug groups, six (75.0%) reported them as *type A* prevalence rates [28,32,34,36,41,43], and five studies (62.5%) reported them as *type B* prevalence rates [27,30,34,36,43], i.e., from three studies (37.5%) both prevalence types could be extracted [34,36,43]. The list containing all drug frequencies is provided in Appendix A.

In total, 13 drug groups were reported at ATC level 1 by any study, 35 groups at ATC level 2, 16 groups at ATC level 3, and 12 groups at ATC level 4. Table 3 ranks type A prevalence rates of implicated drug groups at ATC level 1 in descending order of estimated point prevalence rates. Corresponding drug groups at ATC level 2 are ranked underneath. Full details of lower hierarchical levels and type B prevalence rates are provided in Appendix A.

At ATC Level 1, three drug groups had substantially higher type A prevalence rates than all others, namely ‘nervous system’ (21.2 [13.2; 32.4]%), ‘cardiovascular system’ (19.9 [10.2; 35.0])%) and ‘blood and blood-forming organs’ (18.0 [13.0; 24.4]%). Type B prevalence rates were similar for nervous system and cardiovascular system but somewhat lower for blood and blood-forming organs.

At ATC level 2, four drug groups had prevalence rates ≥ 10%, namely ‘antithrombotic agents’ (18.2 [11.0; 28.6]%), ‘agents acting on the renin-angiotensin system’ (11.6 [7.2; 18.1]%), ‘diuretics’ (11.4 [7.2; 17.6]%), and ‘immunosuppressants’ (10.6 [4.5; 23.1]%). Type B prevalence rates were similar for antithrombotic agents (13.1 [8.9; 19.0]%), agents acting on the renin-angiotensin system (12.3 [9.1; 16.6]%), and diuretics (11.2 [7.3; 16.7]%) but lower for immunosuppressants (2.6 [0.3; 16.7]%).

## 4. Discussion

### 4.1. Summary of Principal Findings

We conducted a systematic review and meta-analyses of drug-related hospital admissions, in order to provide up-to-date estimates of their prevalence and preventability, as well as the ADRs and drugs implicated, based on studies published in the last ten years. While a number of reviews have previously been published, regular updates are essential to inform future improvement efforts, given that medication safety in primary care is receiving increasing attention [1], potentially affecting both the scale and the nature of the problem.

In meta-analysis (twelve studies), the prevalence of ADR-related admissions to ED or inpatient wards was 8.3 [6.4–10.7]%, while the corresponding figure for ADEs (five studies) was somewhat higher at 13.9 [8.1; 22.8]%. Almost half (44.7 [28.1; 62.4]%) of ADR-related admissions and more than two-thirds (71.0 [65.9; 75.6]%) of ADE-related admissions were estimated to be at least potentially preventable. This may be explained by the fact that ADEs (unlike ADRs), by definition, always also include inappropriate use of medication, which is generally more likely to be avoidable.

In further meta-analyses, the ADR categories most frequently implicated in ADR-related admissions (point estimates ≥10.0%) were gastrointestinal disorders, electrolyte disturbances, bleeding events, and renal and urinary disorders. The drug groups with the highest reported prevalence rates were ‘nervous system’, ‘cardiovascular system’, and ‘blood and blood-forming organs’. At ATC level 2, antithrombotic agents were found to be the most commonly implicated drug group, followed by agents acting on the renin-angiotensin system, diuretics, and immunosuppressants.

Comparing the ADRs and drugs most commonly implicated in ADR-related admissions, the fact that bleeding events and antithrombotic drugs both feature among the most commonly reported ADRs and drugs, respectively, provides a consistent message. The same applies to renal and urinary disorders (particularly acute kidney injury) and renin-angiotensin system inhibitors and diuretics.

### 4.2. Comparison with Other Literature

Numerous previous systematic reviews have been conducted on the topic of drug-related hospital admissions, but only a few have conducted meta-analyses of prevalence rates, and none (to the best of our knowledge) have conducted meta-analyses of contributing ADRs and drugs. In addition, differences in study aims and therefore inclusion and exclusion criteria (e.g., elderly patients or preventable ADRs only) limit the comparability to the findings of our review.

*ADR and ADE prevalence.* An earlier systematic review of *ADR*-related hospital admissions published in 2008 [9] found a median (IQR) ADR prevalence rate of 6.3 (3.9; 9.0)% in younger patients and 10.7 (9.6; 13.3)% in older patients. A more recent systematic review published in 2014 by Al Hamid et al. reported a median prevalence rate of 7.0 (IQR, 2.4–14.9)% [7]. In comparison, our meta-analyses found slightly higher prevalence rates between 7.3 [95% CI 5.7–9.5]% and 9.3 [6.4–13.3]% depending on whether ED visits and admissions related to ADRs were included. In contrast, our meta-analysis found the prevalence rate of *ADE*-related hospital admissions (12.0 [5.8–23.1]%) based on five studies published between 2015 and 2021 to be substantially higher than the prevalence rate of 4.6% (IQR, 2.85–16.6) reported by Al Hamid et al. based on two retrospective and four prospective studies published between 2001 and 2009 [7]. However, restriction to prospective studies (as in our meta-analysis) yields a prevalence rate of 12.4% (IQR, 3.75–22.9), which is comparable to our estimate.

*ADR preventability.* A meta-analysis by Hakkarainen et al. found an ADR preventability rate of 52 (42–62)% [53] based on 16 studies published between 1994 and 2008, while a meta-analysis by Patel et al. found a preventability rate of 45 (33.1–57.2)% based on 22 studies published between 2001 and 2015 [54] In comparison, we found a very similar ADR preventability rate of 44.7 (28.1–62.4)% based on four studies published between 2014 to 2021.

*Implicated ADRs.* In their systematic review of 43 studies published between 2000 and 2015, Angamo et al. reported that the three most commonly reported ADRs were gastrointestinal (GI) bleeding, electrolyte and metabolic disturbances, and cardiovascular disorders [55]. However, the authors did not quantify these statements. Nevertheless, the findings of our meta-analysis were similar, in that we identified gastrointestinal disorders, electrolyte disturbances, bleeding events, and renal and urinary disorders as the most commonly implicated ADRs.

*Implicated drug groups.* In our meta-analysis, ‘nervous system’ (21.2%), ‘cardiovascular system’ (19.9%), and ‘blood and blood-forming organs’ (18.0%) (especially antithrombotics) were found to be most commonly implicated in ADR-related admissions. In the systematic review by Kongkaew et al., central nervous system drugs accounted for 9.7% in younger and 13.8% in older adults, while cardiovascular drugs and antithrombotics combined accounted for 45.7% and 42.5% in younger and older people, respectively [9]. In the review by Kongkaew et al., nonsteroidal anti-inflammatory drugs (NSAIDs) featured among the main medications implicated in ADR-related hospital admissions (14.6% and 18.8% in younger and older adults, respectively) [9]), whereas the prevalence in our review was considerably lower at 6.2% (95% CI, 3.1–11.9). It is possible that increased attention to the cardiovascular, renal, and gastrointestinal risks of NSAIDs, e.g., through their consistent inclusion in lists of potentially inappropriate medication [56,57] and publications of successful improvement interventions [58,59], have led to a more cautious use of these drugs over the last decade.

In contrast, our findings could suggest that the overall burden of ADR-related hospital admissions may have increased. Nevertheless, such longitudinal comparisons should be interpreted with caution given that differences in findings could also be attributed to heterogeneity in terms of geographical setting and methods.

### 4.3. Strengths and Limitations

A key strength of this study is that it provides not only an up-to-date assessment of the magnitude of the problem of drug-related hospital admissions but also of their preventability and the drugs, ADRs and ADEs implicated. It is one among only a few systematic reviews on this topic, which have conducted meta-analyses on the prevalence of drug-related admissions, and to the best of our knowledge, it is the first to have conducted meta-analyses on ADRs and drugs implicated.

Meta-analyses are challenging in this field because of the heterogeneity of study populations, definitions of ADEs/ADRs and their measurement, the classification of drugs and ADEs/ADRs, and the reporting of prevalence rates. We accounted for this heterogeneity by excluding studies restricted to particular populations (i.e., age or admitted to specific wards), those without a standardized causality assessment, and those using data sources other than medical chart review to identify ADRs/ADEs. In addition, we stratified analyses according to whether ADRs or ADEs were reported and according to how prevalence rates were reported. We also conducted a number of sensitivity analyses in order to assess potential bias due to the inclusion/exclusion of ED visits that did not lead to admissions to inpatient wards, due to the use of predefined lists for ADR identification (trigger tools), and due to differently reported causal links between ADRs/ADEs and hospital admissions (i.e., “related”, “causing”, “contributing”).

Nevertheless, our aim to provide up-to-date estimates of prevalence rates and our efforts to minimize heterogeneity limited the number of studies that could be included in the meta-analyses. Despite our best efforts to minimize study heterogeneity, residual heterogeneity remained, including different definitions of ADRs and ADEs, prospective vs. retrospective study designs, and the use of different causality assessment methods. The reporting in the studies often left room for interpretation as to which events were included in drug-related admissions and there was usually no formal classification of the strength of the causal link between ADRs/ADEs and hospital admissions. In addition, the drugs and especially the ADRs/ADEs were often not classified in a standardized way (ATC or ICD-10/MedDRA). Identifying the most important contributors to drug-related admission is inherently difficult, as prevalence rankings are vulnerable to how individual ADRs are grouped or categorized at higher levels. For example, gastrointestinal bleeding could be attributed to both gastrointestinal disorders and bleeding events, affecting the prevalence of these higher-level categories. Despite these uncertainties, the fact that prevalence rates of ADRs (and ADEs albeit to a lesser extent) were relatively robust in sensitivity analysis, and that findings of the most frequently implicated ADRs and drugs provided consistent messages, increases the confidence in our findings.

### 4.4. Implications for Research and Practice

Consistent with earlier systematic reviews on this topic [7,9,53,54,60], we found substantial residual heterogeneity between prevalence rates reported by included studies. Wallerstedt et al. recently highlighted methodological sources of such heterogeneity, criticizing that ADRs with only a *possible* relationship to drug treatment are often included, and that it is often unclear whether ADRs simply *contributing* to admissions rather than *causing* them are included [14]. However, many adverse events are multi-causal and even though drug therapy may thus often not be the only cause, it may still be decisive for its occurrence. While more exclusive definitions may therefore overestimate the scale of the problem of drug-related hospital admissions, more restrictive definitions may well underestimate it. We therefore suggest that future studies should opt for broad definitions but consistently differentiate between ADRs and ADEs, between the potential, probable, and certain drug-relatedness of events, between drug-related events causing vs. contributing to admissions, and between emergency department visits with and without subsequent admissions to inpatient wards. For better comparability of the frequencies of implicated ADRs/ADEs and drugs, especially to improve the feasibility of meta-analyses, their classification should follow a uniform coding system according to ICD-10 or MedDRA for diagnoses and according to the ATC classification system for drugs, respectively, with the results being published in this way.

Drug-related hospital admissions are in many ways ideal measures for identifying opportunities for improving the safety of medication use in primary care and for monitoring changes over time. However, most included studies are single center studies with relatively small sample sizes, which makes them vulnerable to selection bias and random error. Further research is required to further develop, validate, and implement less resource-intensive approaches to enable repeated measurement of the prevalence and nature of drug-related hospital admissions over time.

Examination of the types of ADRs and drugs implicated in drug-related admissions can enable prioritization of primary care processes to be targeted by medication safety interventions. Our findings suggest that targeting gastrointestinal disorders, bleeding events and renal/urinary disorders as well as those linked to the nervous system, cardiovascular system and antithrombotic agents may yield the largest impact. While generic primary care interventions targeting medication safety (including the appropriate use of polypharmacy) in its entirety have had limited effects, there is some evidence that interventions targeting the high-risk use of specific drug groups, such as NSAIDs, antiplatelets, and benzodiazepines, can improve care processes and outcomes [58,59,61]. More research is required to identify intervention components to address medication specific barriers to the safe use of medicines, especially for nervous system and cardiovascular system drugs.

## 5. Conclusions

The up-to-date estimates resulting from our systematic review and meta-analysis confirm that drug-related admissions to the ED and inpatient wards remain a major health care problem, much of which is avoidable. Our findings suggest that cardiovascular and antithrombotic agents remain important targets for intervention and that the contribution of nervous system drugs to the overall burden of ADR-related hospital admissions has increased while the contribution of NSAIDs may have declined. Nevertheless, these findings should be interpreted with caution due to the heterogeneity of included studies. Further research is required to enable robust, resource-efficient, and repeated measurement of drug-related hospital admissions in order to monitor local changes in primary-care medication safety over time.

## Figures and Tables

**Figure 1 jcm-12-01320-f001:**
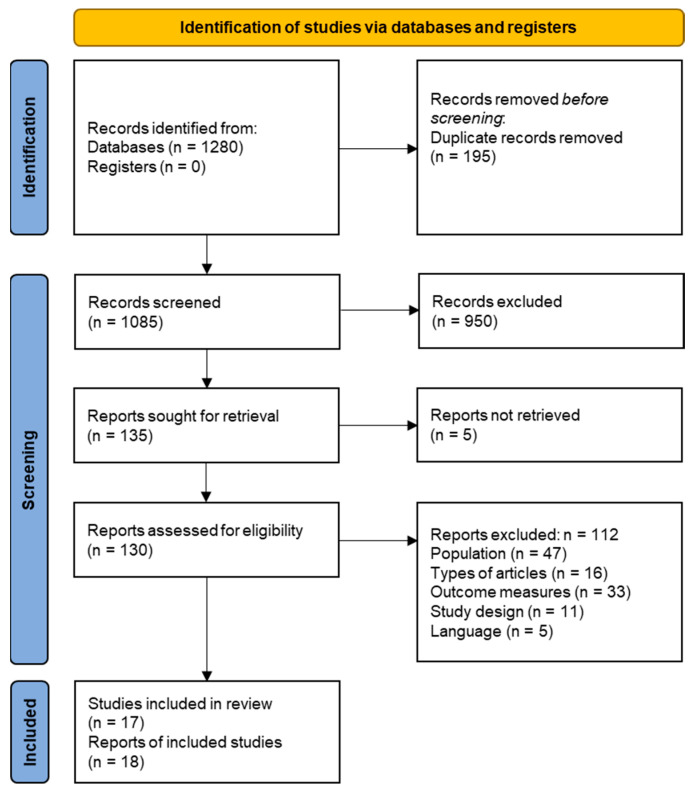
PRISMA flow chart showing the selection process of included studies.

**Figure 2 jcm-12-01320-f002:**
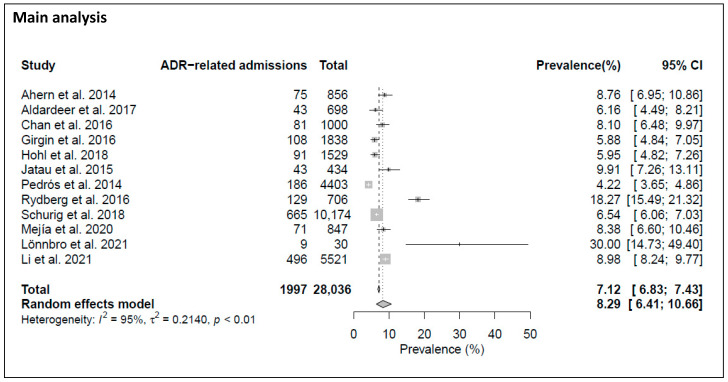
Main analysis. Forest plot of the pooled prevalence of ADR-related admissions to EDs or inpatient wards [5,27,28,29,30,31,32,37,38,40,41,43].

**Figure 3 jcm-12-01320-f003:**
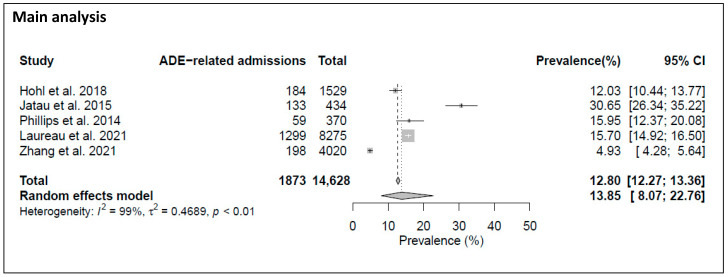
Forest plot of the pooled prevalence of ADE-related admissions to EDs or inpatient wards [35,39,40,42,43].

**Figure 4 jcm-12-01320-f004:**
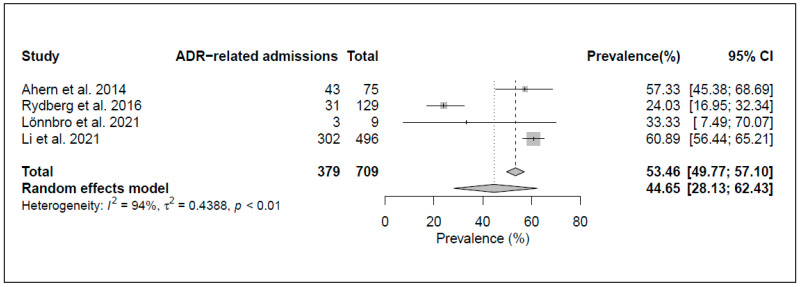
Forest plot of the pooled preventability rates of ADR-related hospital admissions [27,28,31,41].

**Figure 5 jcm-12-01320-f005:**
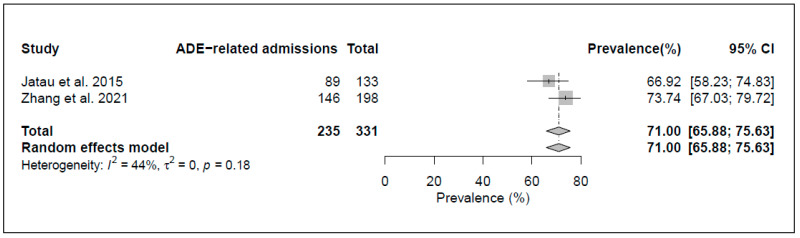
Forest plot of the pooled preventability rates of ADE-related ED visits/hospital admissions [39,40].

**Table 1 jcm-12-01320-t001:** Main characteristics of the selected studies on ED visits and/or hospital admissions related to ADRs or ADEs.

Author, Year	Location	Data Collection (P/R)	Duration (Months)	Sample Size (n)	Age (Years)	Age (years) Mean (SD)/ * Median (IQR or Range)	Gender n Female (%)	ADR/ADE Definition	Causality Method	Preventability Method Rate		ADR or ADE-Related / All Admissions % [95 % CI]
**Adverse drug reactions (ADRs)**												
**ADR-related hospital admissions (through the ED)**												
Ahern et al., 2014 [27]	Ireland	P	1	856	all ages	ADR: 68.8 (18.0) No ADR: 44.8 (25.8)	ADR: 37 (49.3) No ADR: 366 (46.9)	WHO	Naranjo	Hallas	43/75 (57.3)	75/856 8.8 [7.0–10.9]
Alayed et al., 2019 [36]	Saudi Arabia	P	6	unclear	> 12	49.1 (22.4)	18 (47.4)	WHO	Naranjo	-	-	38/-
Aldardeer et al., 2017 [37]	Saudi Arabia	R	6	698	all ages	MREA: 55 No MREA: 54.3	326 (46.7)	Author defined	Naranjo	-	-	43/698 6.2 [4.5–8.2]
Chan et al., 2016 [38]	Singapore	P	3	1000	≥ 21	62.8 (16.9)	474 (47.4)	Edwards/ Aronson	LCAT	Hallas	81/83 (97.6)	81/1000 8.1 [6.5–10.0]
Li et al., 2021 [41]	Australia	R	3	5521	> 18	* ADR: 74 (IQR: 58–83) No ADR: 55 (IQR: 35–72)	ADR: 264 (53.2) No ADR: 2760 (54.9)	Not reported	WHO-UMC	Hallas	302/496 (60.9)	496/5521 9.0 [8.2–9.8]
Lönnbro et al., 2021 [28]	Sweden	R	0.5	30	≥ 18	* 72 (range: 25–93)	15 (50.0)	Not reported	WHO-UMC	Hallas	3/9 (33.3)	9/30 30.0 [14.7–49.4]
Mejía et al., 2020 [29] ^△^	Spain	R	1	847	all ages	* 75 (range: 26–100)	46 (48.4)	Spanish Ministry of Health	SPhVS Algorithm	-	-	71/847 8.4 [6.5–10.3]
Pedrós et al., 2014 [30] ^△^	Spain	P	4	4403	all ages	* ADR: 75 (range: 28–97) No ADR: 66 (range: 16–102)	ADR: 82 (44.1) No ADR: 1660 (39.4)	EU	SPhVS Algorithm	-	-	186/4403 4.2 [3.7–4.8]
Rydberg et al., 2016 [31]	Sweden	P	13	706	≥ 18	* 71 (IQR: 58–82)	351 (49.7)	Nebeker	Naranjo	Hallas	31/129 (24.0)	129/706 18.3 [15.5–21.3]
**ADR-related ED visits and hospital admissions**												
Girgin et al., 2016 [32]	Turkey	P	1	1838	≥ 17	51.5	59 (54.6)	WHO	WHO-UMC	-	-	108/1838 5.9 [4.8–7.1]
Hohl et al., 2018 [43]	Canada	P	12	1529	≥ 19	59.3 (20.9)	851 (55.7)	WHO	Naranjo (adapted)	-	-	91/1529 6.0 [4.8–7.3]
Jatau et al., 2015 [40]	Malaysia	P	1.5	434	all ages	41.0 (21.6) Age distribution: ≤ 19 (6.0), ≥ 20 (94.0)	83 (62.0)	WHO	French method (Queneau et al. [52])	Nelson and Talbert criteria	Not reported	43/434 9.9 [7.3–13.1]
Just et al., 2020 [33,34]	Germany	P	27	NA	≥ 18	* 73 (IQR: 58–80)	1100 (49.7)	ICH	WHO-UMC	-	-	2215/-
Schurig et al. 2018 [5]	Germany	P	1	10174	≥ 18	* 74.5 (range: 18–97)	55 %	EMA	WHO-UMC	(Schumock and Thornton)	Not reported	665/10174 6.5 [6.1–7.0]
**Adverse drug events (ADEs)**												
**ADE-related hospital admissions**												
Phillips et al., 2014 [42]	Australia	P	2	370	all ages	* 64 (IQR: 46–80)	171 (46.2)	American Society of Health-System Pharmacists	Jones‘ algorithm	Chan et al. [50]	47/72 (65.3)	59/370 16.0 [12.4–20.1]
Zhang et al., 2021 [39]	China	P	33	4020	all ages	Age distribution: ≤ 39 (4.0), 40–69 (28.3), ≥ 70 (67.7)	99 (50.0)	Not reported	Naranjo	Schumock and Thornton	146/198 (73.7)	198/4020 4.9 [4.3–5.6]
**ADE-related ED visits and hospital admissions**												
Hohl et al., 2018 [43]	Canada	P	12	1529	≥ 19	59.3 (20.9)	851 (55.7)	Author defined	Naranjo (adapted)	-	-	184/1529 12.0 [10.4–13.8]
Jatau et al., 2015 [40]	Malaysia	P	1.5	434	all ages	41.0 (21.6) Age distribution: ≤ 19 (6.0), ≥ 20 (94.0)	83 (62.0)	Author defined	French method (Queneau et al. [52])	Nelson and Talbert criteria	89/133 (66.9)	133/434 30.7 [26.3–35.2]
Laureau et al., 2021 [35]	France	P	6	8275	> 18	59.7 (22.9)	4282 (51.7)	Nebeker	French method and Naranjo (modified)	-	-	1299/8275 15.7 [14.9–16.5]

Abbreviations: ADR = adverse drug reaction; ADE = adverse drug event; ED = emergency department; MR(E)A = medication-related (emergency) admission; P = prospective; R = retrospective; SD = standard deviation; CI = confidence interval; IQR = interquartile range; WHO(-UMC) = World Health Organization(-Uppsala Monitoring Centre); LCAT = Liverpool ADR Causality Assessment Tool; SPhVS = Spanish Pharmacovigilance System (modification of the algorithm of Karch and Lasagna); EU = European Union; EMA = European Medicines Agency; ICH = International Conference on Harmonisation. * Median (IQR or range). ^△^ These studies used a pre-defined list of diseases or syndromes potentially caused by drugs for patient selection.

**Table 2 jcm-12-01320-t002:** ADRs implicated in ADR-related admissions in hierarchical descending order of estimated point prevalence. The X marks studies contributing to the corresponding meta-analysis.

ADR Categories	ADR Frequency as a Proportion of *All Cases with ADRs*			
	Girgin et al. 2016 [32]	Hohl et al. 2018 [43]	Lönnbro et al. 2021 [28]	Prevalence % [95% CI]
Gastrointestinal disorders	X	X	X	**28.5 [21.6; 36.6]**
Electrolyte disturbances		X		**16.5 [9.5; 25.7]**
Bleeding	X	X	X	**13.5 [9.5; 18.8]**
Renal and urinary disorders	X	X	X	**11.8 [5.4; 24.1]**
Skin reactions	X	X		**9.7 [2.2; 33.9]**
Cardiac and vascular disorders	X	X	X	**8.7 [5.5; 13.3]**
Infection		X	X	**5.7 [1.0; 27.4]**
Nervous system disorders	X	X		**5.5 [3.1; 9.7]**
Blood dyscrasias		X	X	**5.0 [2.1; 11.5]**
Musculoskeletal disorders		X		**4.4 [1.2; 10.9]**
Metabolism and nutrition disorders		X		**2.2 [0.3; 7.7]**
Liver disorders		X		**1.1 [0.0; 6.0]**

Abbreviations: ADR = adverse drug reaction; CI = confidence interval.

**Table 3 jcm-12-01320-t003:** Drug groups implicated in ADR-related admissions in hierarchical descending order of estimated point prevalence. The X marks studies contributing to the corresponding meta-analysis.

ATC Code	Drug Groups	Drug Frequency as a Proportion of *All Cases with ADRs*						
ATC Level 1		Alayed et al. 2019 [36]	Girgin et al. 2016 [32]	Hohl et al. 2018 [43]	Just et al. 2020 [34]	Li et al. 2021 [41]	Lönnbro et al. 2021 [28]	Prevalence % [95% CI]
ATC Level 2								
**N**	**Nervous system**		X	X		X	X	**21.2 [13.2; 32.4]**
N02	Analgesics			X			X	9.0 [4.8; 16.4]
N05	Psycholeptics			X			X	7.0 [3.4; 14.0]
N03	Antiepileptics	X		X	X			3.3 [2.7; 4.1]
N06	Psychoanaleptics			X				3.3 [0.7; 9.3]
N07	Other nervous system drugs			X				2.2 [0.3; 7.7]
N04	Anti-parkinson drugs	X			X			1.6 [1.2; 2.2]
**C**	**Cardiovascular system**		X	X		X	X	**19.9 [10.2; 35.0]**
C09	Agents acting on the renin-angiotensin system	X		X			X	11.6 [7.2; 18.1]
C03	Diuretics	X		X	X		X	11.4 [7.2; 17.6]
C07	Beta blocking agents	X		X	X		X	5.7 [2.1; 14.6]
C08	Calcium channel blockers			X	X			3.9 [3.2; 4.7]
C10	Lipid modifying agents				X			1.2 [0.8; 1.7]
**B**	**Blood and blood forming organs**		X	X		X	X	**18.0 [13.0; 24.4]**
B01	Antithrombotic agents	X		X	X		X	18.2 [11.0; 28.6]
B03	Antianemic preparations			X	X			0.2 [0.1; 0.5]
**L**	**Antineoplastic and immunomodulating agents**			X	X	X	X	**9.7 [5.4; 16.6]**
L04	Immunosuppressants	X					X	10.6 [4.5; 23.1]
L01	Antineoplastic agents	X		X			X	5.8 [3.0; 11.2]
**J**	**Antiinfectives for systemic use**		X	X		X		**9.4 [2.0; 35.2]**
J01	Antibacterials for systemic use	X		X	X			7.2 [3.9; 12.9]
J02	Antimycotics for systemic use		X					1.1 [0.0; 6.0]
**A**	**Alimentary tract and metabolism**		X	X		X	X	**8.7 [4.4; 16.3]**
A10	Drugs used in diabetes	X		X	X			2.7 [2.1; 3.4]
A02	Drugs for acid related disorders				X			1.9 [1.3; 2.5]
A06	Drugs for constipation				X		X	0.8 [0.5; 1.3]
A11	Vitamins			X	X			0.4 [0.2; 0.8]
A12	Mineral supplements				X			0.3 [0.1; 0.6]
**M**	**Musculo-skeletal system**		X	X		X		**4.8 [2.0; 11.0]**
M01	Antiinflammatory and antirheumatic products	X		X				6.2 [3.1; 11.9]
M03	Muscle relaxants			X				1.1 [0.0; 6.0]
M04	Antigout preparations				X			0.5 [0.2; 0.8]
**V**	**Various**		X	X				**3.0 [1.4; 6.6]**
V08	Contrast media			X				2.2 [0.3; 7.7]
**H**	**Systemic hormonal preparations**		X	X		X		**1.9 [1.1; 3.2]**
H02	Corticosteroids for systemic use			X	X			3.8 [3.1; 4.7]
H01	Pituitary and hypothalamic hormones and analogues			X				1.1 [0.0; 6.0]
H03	Thyroid therapy				X			0.9 [0.5; 1.3]
**R**	**Respiratory system**		X	X		X		**1.5 [0.4; 5.0]**
R01	Nasal preparations			X				1.1 [0.0; 6.0]
R05	Cough and cold preparations			X				1.1 [0.0; 6.0]
R06	Antihistamines for systemic use			X				1.1 [0.0; 6.0]
R03	Drugs for obstructive airway diseases				X			1.0 [0.6; 1.4]
**P**	**Antiparasitic products, insecticides and repellents**			X				**1.1 [0.0; 6.0]**
**D**	**Dermatologicals**		X					**0.9 [0.0; 5.1]**
**G**	**Genito urinary system and sex hormones**		X	X		X		**0.7 [0.3; 1.7]**
G04	Urologicals			X	X			0.6 [0.4; 1.0]

Abbreviations: ATC = Anatomical Therapeutic Chemical (classification system); CI = confidence interval.

## Data Availability

The data presented in this study are openly available in this article and in the Appendix A.

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
