# Peer review of "Which Adverse Events and Which Drugs Are Implicated in Drug-Related Hospital Admissions? A Systematic Review and Meta-Analysis"

_jcm, 2023, doi:10.3390/jcm12041320_

Round 1

Reviewer 1 Report

I reviewed with interest this manuscript. This is up-to-date metaanalysis is useful regarding the aim of reduction of medications harms.

To my point-of-view, the work could be clearer if the authors include ADE into ADR. They could them merge figures 2 and 3, and figure 4 and 5. This is confusing, particularly in the abstract, and frontiers between direct or indirect relatedness is not so clear. Then, the title could be: "which adverse reactions..."

As the 3 papers used in ADR frequency metaanalysis are included in Drug frequency metaanalysis, could a link between ADR/drug could be discussed?

Minor comment: error in date of Laureau et al. : 2021 (not 2020)

Reviewer 2 Report

General comments

·       Why numbering in the Abstract section? Better to rearrange keywords in alphabetic order.

Specific comments

·       You’ve cited supplementary files, but I could not find any of these files. These could not even be downloaded from the web link www.mdpi.com/xxx/s1 that you provided.

Results

·     Table 1: You’ve categorized study design as prospective or retrospective under the ‘Study design’ column but the prospective or retrospective nature just reflects the direction of enquiry, rather than reflecting the study design.

·       Table 3: Rewrite “Pituitary and hypothalamic hormones and ana-“ completely.

References

·       References 4, 13, 15, 17, 18, 20, 26, 35, 61: Rewrite name of the journals properly.

·       References 21, 44: Write the name of the journals.

·       Remove the typo in the reference 52 and its citation.
